# Ultrasound Imaging of Facial Vascular Neural Structures and Relevance to Aesthetic Injections: A Pictorial Essay

**DOI:** 10.3390/diagnostics12071766

**Published:** 2022-07-21

**Authors:** Wei-Ting Wu, Ke-Vin Chang, Hsiang-Chi Chang, Chen-Hsiang Kuan, Lan-Rong Chen, Kamal Mezian, Vincenzo Ricci, Levent Özçakar

**Affiliations:** 1Department of Physical Medicine and Rehabilitation, National Taiwan University Hospital, Bei-Hu Branch, Taipei 10845, Taiwan; wwtaustin@yahoo.com.tw (W.-T.W.); lchen@livemail.tw (L.-R.C.); 2Department of Physical Medicine and Rehabilitation, College of Medicine, National Taiwan University, Taipei 10048, Taiwan; 3Center for Regional Anesthesia and Pain Medicine, Wang-Fang Hospital, Taipei Medical University, Taipei 11600, Taiwan; 4Department of Physical Medicine and Rehabilitation, Taichung Veterans General Hospital, Taichung 407219, Taiwan; s19801041@gm.ym.edu.tw; 5Department of Surgery, Division of Plastic Surgery, National Taiwan University Hospital, Taipei 10048, Taiwan; chkuan0408@gmail.com; 6Department of Rehabilitation Medicine, Charles University, First Faculty of Medicine and General University Hospital in Prague, 12800 Prague, Czech Republic; kamal.mezian@gmail.com; 7Physical and Rehabilitation Medicine Unit, Department of Biomedical and Neuromotor Science, Istituto di Ricovero e Cura a Carattere Scientifico Rizzoli Orthopedic Institute, 40136 Bologna, Italy; vincenzo.ricci58@gmail.com; 8Department of Physical and Rehabilitation Medicine, Hacettepe University Medical School, Ankara 06100, Turkey; lozcakar@yahoo.com

**Keywords:** sonography, injection, safety, rejuvenation, cosmetics

## Abstract

The facial and submental regions are supplied by complicated neurovascular networks; therefore, facial aesthetic injections may be associated with serious adverse events such as skin necrosis and blindness. Pre-injection localization of neurovascular structures using high-resolution ultrasound can theoretically prevent unexpected complications. Therefore, a systematic protocol that focuses on these facial neurovascular structures is warranted. In this pictorial essay, we discuss the sonoanatomy of facial and submental neurovascular structures and its relevance to aesthetic injections. Moreover, we have highlighted the mechanisms underlying potential neurovascular injuries during aesthetic injections.

## 1. Introduction

Facial aesthetic injections are widely used in clinical practice; the most common indications include reduction of wrinkles and facial contouring. Injectables include botulinum toxin [1], fillers (hyaluronic acid, silicone, and autologous fat) [2] and deoxynucleic acid [3]. Botulinum toxin blocks the neuromuscular junction, which results in relaxation of the underlying muscles and is therefore used to eliminate wrinkles. Fillers used for volume augmentation are useful for contouring atrophic/concave regions. Deoxynucleic acid is useful to reduce local fullness in areas of excessive fat accumulation. However, the facial and submental regions are supplied by a complicated neurovascular network; therefore, post-injection complications represent a significant challenge to cosmetologists [4,5].

Ultrasonography is known to provide high-resolution imaging of superficial soft tissues [6,7]. Power Doppler imaging enables investigators to scrutinize even tiny vasculature without the administration of contrast agents [8]. Although sound beams cannot penetrate the bony cortex, ultrasonography is a sensitive tool to delineate the neurovascular foramina on facial bones [7]. A recent review discussed details of the sonoanatomy of facial muscles but not of facial neurovascular structures [9]. In this pictorial essay, we discuss the sonoanatomy of the neurovascular structures in the facial and submental regions and its relevance to aesthetic injections, aiming to facilitate pre-procedure ultrasound scanning in a systematic manner. We hope the accidental injury to the vessels and nerves on the face can thus be decreased during aesthetic injections. Figure 1, Figure 2 and Figure 3 illustrate the main neurovascular structures over the frontal, lateral, and submental regions of the face, respectively. These figures have been adapted to the face of our co-author (H.-C. C) with permission obtained for publication. All sonographic images were obtained using a 10–25 MHz high-frequency linear transducer (X-Cube 90, Alpinion Medical Systems Co. Ltd., Anyang, Korea).

## 2. Aesthetic Injections and Relevant Neurovascular Structures

### 2.1. Upper Face

Facial wrinkles are the most common indication for botulinum toxin injections [10]. Horizontal forehead lines can be corrected by injections into the frontalis muscle. Likewise, injections into the corrugator supercilii and procerus are useful to eliminate glabellar frown lines. Lateral canthal lines, commonly referred to as ‘crow’s feet’, can be reduced by injections into the lateral orbital portions of the orbicularis oculi muscle.

The forehead and glabella represent the most common sites for filler injections [11]. The injectables should be introduced within the loose connective tissue above the periosteum for wrinkle reduction and volume expansion. Furthermore, injections into the periorbital area can be used for correction of ‘sunken eyes.’

Vessels that course across the middle of the forehead include the supratrochlear artery/vein and the supraorbital artery/vein [12,13]. These are accompanied by the supraorbital and supratrochlear nerves [14]. The lateral forehead is supplied by the frontal branch of the superficial temporal artery and vein [15] (Figure 1).

### 2.2. Middle Face

Temporalis muscle hypertrophy may accompany masseter muscle atrophy, and botulinum injections are useful in such cases. Injections into the risorius and zygomaticus major muscles are occasionally used to improve facial asymmetry. Injections into the levator labii superioris alaeque nasi and levator labii superioris are shown to be useful to correct a gummy smile [9].

Filler injections into the temporal area are useful to correct atrophy secondary to age-induced fat loss [11]. Medial infraorbital region injections are used to trim the tear trough, which is a deep crease between the lower eyelid and upper cheek. Flattening of the malar eminence can be corrected by injections into the cheek, which is referred to as malar augmentation. Injections into the buccal fat pad layer are useful to contour anterior sunken cheeks. Subzygomatic depression can be corrected through injections into the superficial fat of the subcutaneous layer above the superficial musculoaponeurotic system distal to the zygomatic arch. Nasolabial fold injections can improve age-induced deepening of nasolabial lines. Injections directed into the layer between the nasalis muscle and the nasal cartilage increase the projection of the nasal bridge.

The dorsal nasal artery/vein and the angular artery/vein run in close proximity to or on the nose [16,17]. The cheek and upper perioral region are supplied by the superior labial artery/vein and the infraorbital artery/vein [18,19], and the superficial temporal and deep temporal arteries/veins supply the temporal area (Figure 2) [20]. These vessels are accompanied by the auriculotemporal, infraorbital, and facial nerves (Figure 2) [14].

### 2.3. Lower Face

Injections into the masseter muscle are used to contour a square face and for treatment of bruxism [21]. Mentalis muscle injections improve a cobblestone chin. Injections into the depressor labii inferioris or depressor anguli oris muscle are used to treat a droopy face [9].

Filler injections administered medial to the depressor anguli oris muscle can reduce prominent marionette lines. Injections over the mental region improve the appearance of an under-projected chin [11].

The inferior labial artery/vein and the facial artery/vein are important vessels of the lower face (Figure 1) [22]. Innervation of the lower face is primarily via the mental nerve [14].

### 2.4. Submental Region

Submental fullness is the most common indication for aesthetic injections into the submental region. Deoxycholic acid is approved by the Food and Drug Administration of the United States for reduction of submental fat [3]. Although these injections are considered safe and effective, adverse reactions such as local tenderness, edema, numbness, bruising, and hematoma have been reported [3]. The submental artery/vein and the hypoglossal nerve are important neurovascular structures that should be protected during submental injections (Figure 3) [23,24].

## 3. Mechanism of Neurovascular Injury

Facial aesthetic injections are associated with risks such as bruising, edema, skin discoloration, and infection. Nerve injuries may occur secondary to direct trauma such as needle piercing and nerve laceration. Based on severity (from mild to severe), peripheral nerve injury is categorized as neurapraxia, axonotmesis, and neurotmesis [25,26]. Mild nerve injury leads to temporary sensory/motor impairment, which usually recovers within 2 to 3 weeks. However, severe injury (for example, nerve transection) can cause long-term anesthesia over the innervated region.

Vascular compromise observed after filler injection [27] is attributable to filler-induced vessel compression, particularly after enlargement of the volume owing to absorption of water nearby. Direct puncture of vessels can lead to antegrade or retrograde flow of fillers and trigger embolic events, which may result in serious complications such as skin necrosis, blindness, and stroke.

## 4. Sonoanatomy of the Neurovascular Structures of the Upper Face

### 4.1. Supratrochlear Artery, Vein, and Nerve

The supratrochlear artery is the terminal branch of the ophthalmic artery [12]. It originates from the supratrochlear foramen, which is a small aperture at the medial edge of the cranial orbital margin of the frontal bone. The transducer is placed over the medial orbital rim in the horizontal plane, and the medial edge of the transducer is pivoted to the medial canthus. The supratrochlear artery is visualized as it emerges from the supratrochlear foramen [28] (Figure 4). The supratrochlear nerve (Figure 5) [29] (which originates from the frontal nerve, which itself is a branch of the ophthalmic nerve) and vein may be visualized in addition to the supratrochlear artery.

### 4.2. Supraorbital Artery, Vein, and Nerve

The supraorbital artery also branches from the ophthalmic artery and emerges from the supraorbital foramen, a small opening near the midpoint of the superior orbital margin in the frontal bone [12]. The transducer is placed over the center of the superior orbital rim. The supraorbital artery originates from the supraorbital foramen (Figure 6), and the supraorbital nerve [30,31] (also a terminal branch of the frontal nerve) and vein are identified along with the supraorbital artery (Figure 7).

### 4.3. Frontal Branch of the Superficial Temporal Artery/Vein

The superficial temporal artery branches into the terminal frontal branch near the level of the tragus [20,32]. The frontal branch travels anteriorly to supply the lateral frontal region. The transducer is placed over the lateral part of the forehead, slightly cranial to the eyebrows. The frontal branches of the superficial temporal artery (and vein) can be visualized coursing over the frontalis muscle (Figure 8). Table 1 summarizes the techniques used to scan the neurovascular structures of the upper face.

## 5. Sonoanatomy of the Neurovascular Structures of the Middle Face

### 5.1. Dorsal Nasal Artery and Vein

The dorsal nasal artery is a terminal branch of the ophthalmic artery [33] and emerges from the medial orbital rim and descends along the nasal septum. A communicating branch, known as the intercanthal artery, may be identified at the nasal bridge. The transducer is placed in the horizontal plane near the medial orbital rim to visualize the short axis of the dorsal nasal artery (and vein) on the procerus, levator labii superioris alaeque nasi, and nasalis muscles (Figure 9) [9,34].

### 5.2. Angular Artery and Vein

The angular artery is the terminal portion of the facial artery and ascends along the nasolabial sulcus to the medial orbital rim [17]. The transducer is placed in the horizontal plane lateral to the ala of the nose, and the angular artery can be identified coursing over the levator labii superioris alaeque nasi and levator labii superioris muscles [35]. The angular vein is seen more lateral than the angular artery (Figure 10).

### 5.3. Superior Labial Artery and Vein

The superior labial artery originates from the facial artery and courses along the cranial edge of the lip [22]. The transducer is placed in the sagittal plane medial to the angle of the mouth. The superior labial artery/vein can be visualized coursing over or under the orbicularis oris muscle (Figure 11) [22,36].

### 5.4. Infraorbital Artery, Vein, and Nerve

The infraorbital artery is the terminal branch of the maxillary artery [19]. The transducer is placed in the horizontal plane slightly distal to the inferior eyelid. The infraorbital artery (Figure 12), vein, and nerve (a branch from the maxillary nerve, Figure 13) can be visualized as they emerge from the infraorbital foramen [37].

### 5.5. Superficial Temporal Artery and Vein

The superficial temporal artery originates from the external carotid artery and further divides into the frontal and parietal branches [15]. The transducer is placed in the horizontal plane cranial to the zygomatic arch. The parietal branch can be identified superficial to the temporal fascia (Figure 14) [38].

### 5.6. Deep Temporal Artery and Vein

The deep temporal artery, which shows an anterior and a posterior branch [39] originates from the maxillary artery, which itself is a branch of the external carotid artery. The transducer is placed in the horizontal plane, cranial to the zygomatic arch, to identify the temporalis muscle. The deep temporal artery can be visualized between the temporalis muscle and the periosteum of the temporal bone (Figure 15).

### 5.7. Auriculotemporal Nerve

The auriculotemporal nerve branches from the mandibular nerve and provides sensory innervation to the jaw, ears, and scalp [14]. The transducer is placed in the horizontal plane slightly anterior to the tragus. The short axis of the auriculotemporal nerve can be observed adjacent to the superior temporal artery (Figure 16) [40].

### 5.8. Facial Nerve

After emerging from the brain stem, the facial nerve travels through the facial canal in the temporal bone [41] and exits the skull bone through the stylomastoid foramen to enter the parotid gland. The transducer is placed in the horizontal plane at the cranial level of the mandibular ramus. The long axis of the facial nerve is observed as a tubular hypoechoic structure within the parotid gland (Figure 17) [42]. Table 2 summarizes the techniques used to scan neurovascular structures of the middle face.

## 6. Sonoanatomy of the Neurovascular Structures of the Lower Face

### 6.1. Facial Artery and Vein

The facial artery, a branch of the external carotid artery [43], courses superficial to the posterior belly of the digastric muscle, the stylohyoid muscle and the submandibular gland before it ascends toward the nasolabial fold after it crosses the caudal border of the mandible. The transducer is placed in the horizontal plane at the midpoint of the mandibular body to visualize the facial artery (and vein) (Figure 18).

### 6.2. Inferior Labial Artery and Vein

The inferior labial artery, which branches from the facial artery at the inferolateral aspect of the angle of the mouth, Ref. [22] courses along the caudal edge of the lower lip between the orbicularis oris and the mucous membrane. The transducer is placed in the sagittal plane over the lower lip to the inferior labial artery (and vein) along the short axis (Figure 19).

### 6.3. Mental Nerve

The mental nerve is a branch of the mandibular nerve that provides sensation to the chin and the lower lip [14]. The transducer is placed in the horizontal plane, caudal to the lower lip, and lateral to the mental protuberance. The mental nerve can be identified as it emerges from the mental foramen (Figure 20).

## 7. Sonoanatomy of the Neurovascular Structures of the Submental Region

### 7.1. Submental Artery

The submental artery branches from the facial artery near the border of the mandibular body and supplies the submandibular and sublingual glands [23]. The transducer is placed in the coronal plane over the submental region, and the submental artery can be visualized coursing superficial to the mylohyoid muscle (Figure 21).

### 7.2. Hypoglossal Nerve

The hypoglossal nerve is the 12th cranial nerve that innervates the extrinsic and intrinsic muscles of the tongue [24]. The transducer is positioned in the coronal plane over the submental area and is moved from the posterior to the anterior aspect. The hypoglossal nerve can be visualized as a hypoechoic circular structure between the hypoglossal and mylohyoid muscles (Figure 22). The lingual artery and vein may accompany the hypoglossal nerve in the submental region. Table 3 summarizes the techniques used to scan the neurovascular structures of the lower face and submental region.

## 8. Factors Associated with Ultrasound-Guided Aesthetic Injections into the Facial and Submental Regions

Although several pilot studies have reported the feasibility of ultrasound-guided facial aesthetic injections, palpation-based interventions remain widely popular among injectors. Factors that limit ultrasound guidance for facial injections include the following: (A) Facial muscles are thin and superficial [9]; therefore, they need to be delineated using transducers with higher frequencies (for example, 20 MHz). (B) The facial region is not a large plain surface, which interferes with the use of an in-plane approach. Therefore, unlike ultrasound-guided injections into peripheral joints, ultrasound guidance for facial cosmetic injections warrants the following strategy, which is different from conventional ultrasound-guided injections:(i)Injectors unfamiliar with real-time guided injections should perform ultrasound imaging using a marking pen to confirm important neurovascular structures. This method facilitates palpation-based injections after removing the transducer and provides a guideline to avoid the marked regions.(ii)Injectors *familiar* with ultrasound-guided techniques should prepare an ultrasound machine equipped with a high-frequency (>20 MHz) transducer, which also has a small footprint (<2.5 cm in width). The out-of-plane approach, which allows multiple injections in one scanning plane, is recommended to shorten the needle pathway under the skin.

## 9. Conclusions

High-resolution ultrasonography enables visualization of facial vessels and nerves. Inadvertent injections into facial neurovascular structures can result in serious adverse effects, which may be prevented by cautious pre-intervention ultrasound scanning. Despite the complicated neurovascular network of the facial region, the systematic protocol presented in this pictorial essay will serve as a guideline for investigators to familiarize themselves with the imaging technique and develop expertise in the aforementioned ultrasound-guided approach to facial aesthetic injections.

## Figures and Tables

**Figure 1 diagnostics-12-01766-f001:**
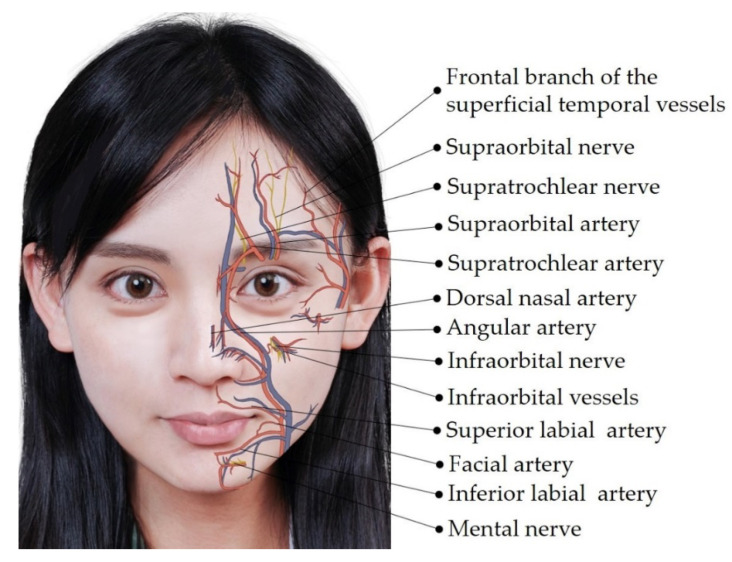
Illustration showing the frontal facial neurovascular structures. Blue structures indicate veins, which are mostly accompanied by arteries (in red) with the same nomenclature.

**Figure 2 diagnostics-12-01766-f002:**
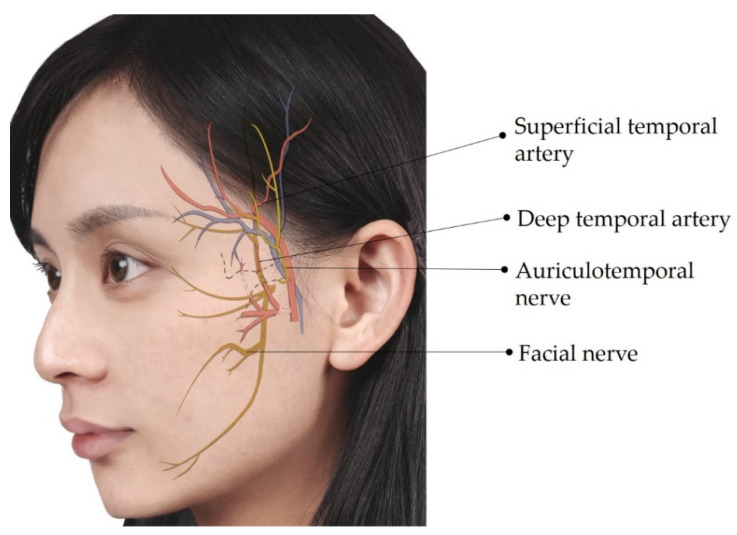
Illustration showing the lateral facial neurovascular structures. Blue structures indicate veins, which are mostly accompanied by arteries (in red) with the same nomenclature.

**Figure 3 diagnostics-12-01766-f003:**
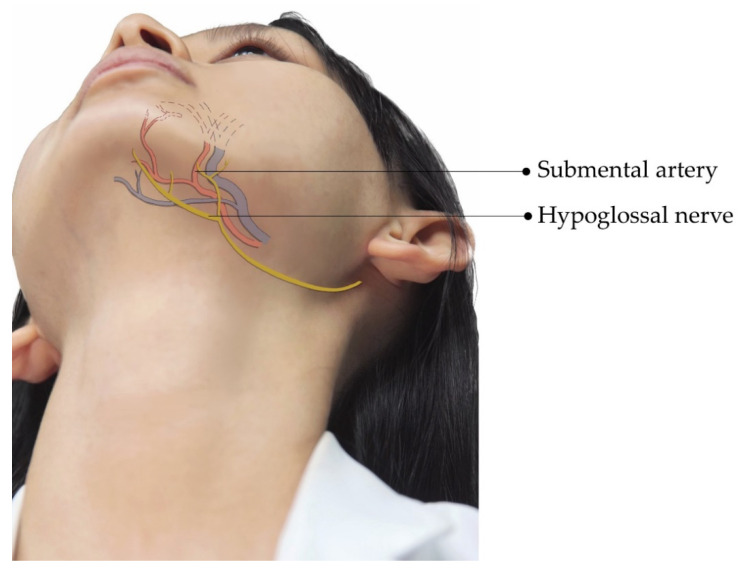
Illustration showing the submental facial neurovascular structures. Blue structures indicate veins, which are mostly accompanied by arteries (in red) with the same nomenclature.

**Figure 4 diagnostics-12-01766-f004:**
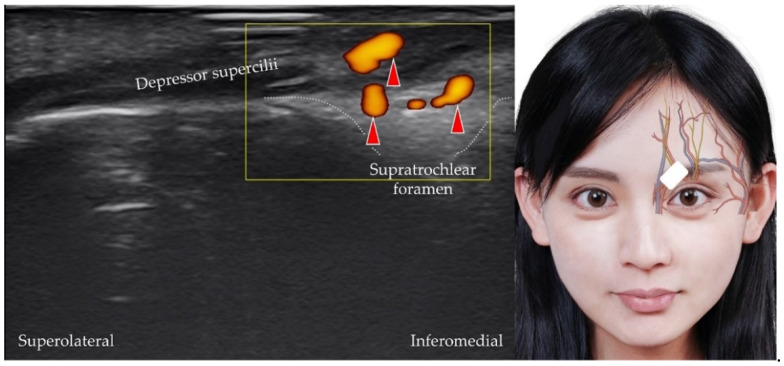
Power Doppler ultrasound imaging of the supratrochlear artery (*red arrowheads*) and its branches. *White square*, footprint of the ultrasound transducer. *White dashed line*, border of the supratrochlear foramen.

**Figure 5 diagnostics-12-01766-f005:**
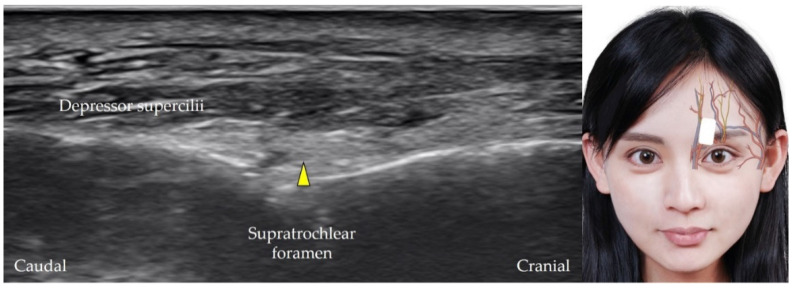
Ultrasound imaging of the supratrochlear nerve (*yellow arrowhead*). *White square*, footprint of the ultrasound transducer.

**Figure 6 diagnostics-12-01766-f006:**
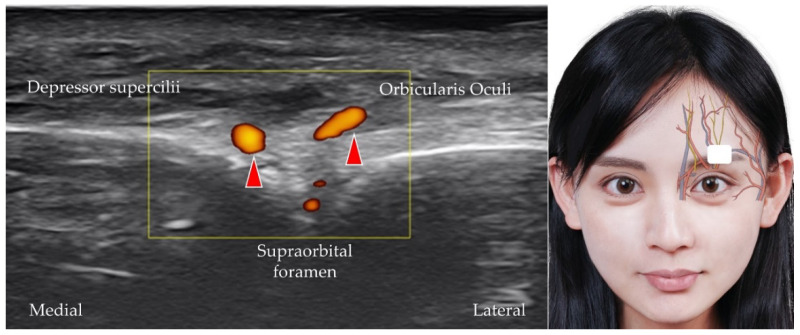
Power Doppler ultrasound imaging of the supraorbital artery (*red arrowheads*) and its branches. *White square*, footprint of the ultrasound transducer.

**Figure 7 diagnostics-12-01766-f007:**
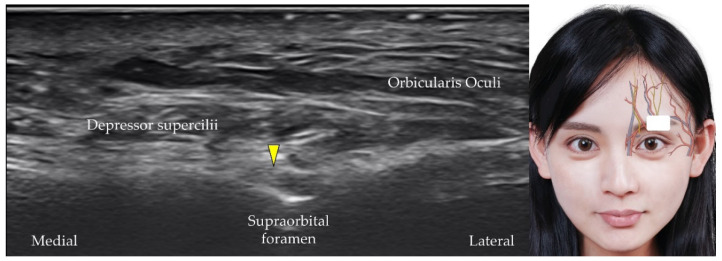
Ultrasound imaging of the supraorbital nerve (*yellow arrowhead*). *White square*, footprint of the ultrasound transducer.

**Figure 8 diagnostics-12-01766-f008:**
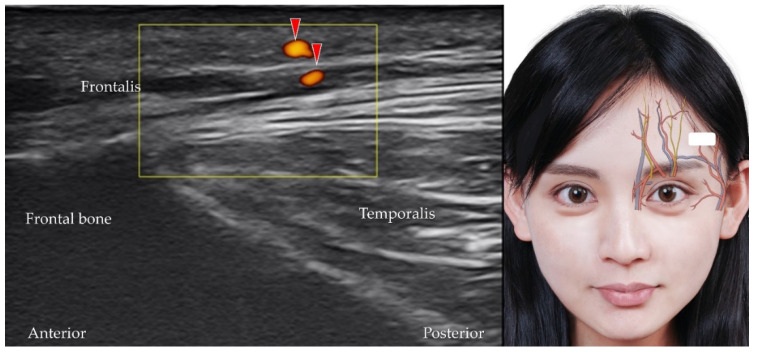
Power Doppler ultrasound imaging of the frontal branch of the superficial temporal artery (*red arrowheads*) and its branches. *White square*, footprint of the ultrasound transducer.

**Figure 9 diagnostics-12-01766-f009:**
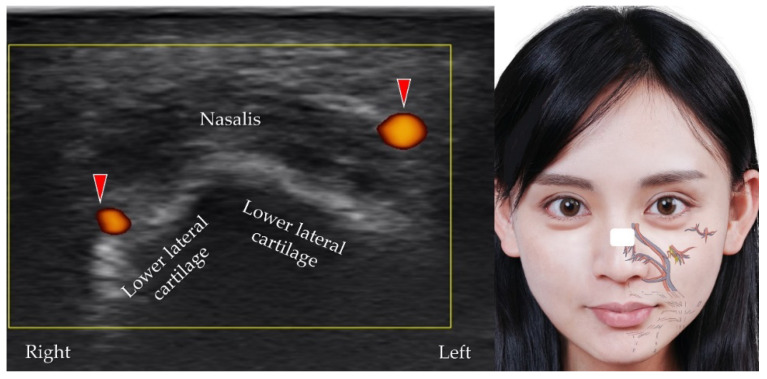
Power Doppler ultrasound imaging of the dorsal nasal artery (*red arrowheads*) and its branches. *White square*, footprint of the ultrasound transducer.

**Figure 10 diagnostics-12-01766-f010:**
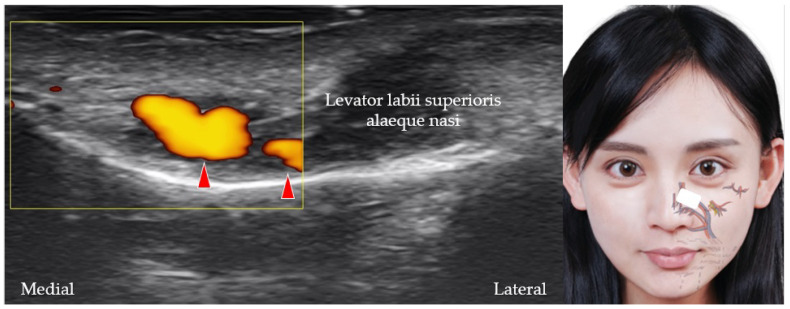
Power Doppler ultrasound imaging of the angular artery (*red arrowheads*). *White square*, footprint of the ultrasound transducer.

**Figure 11 diagnostics-12-01766-f011:**
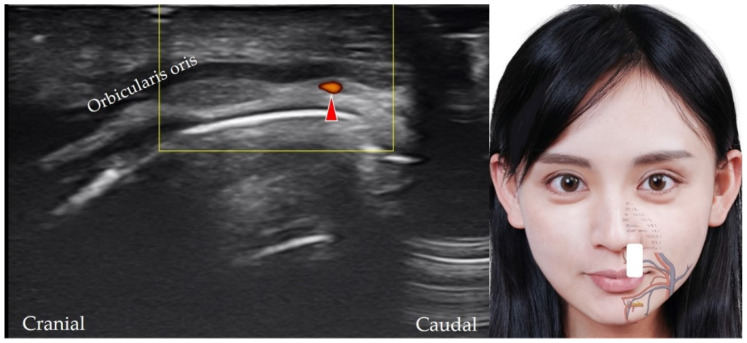
Power Doppler ultrasound imaging of the superior labial artery (*red arrowheads*). *White square*, footprint of the ultrasound transducer.

**Figure 12 diagnostics-12-01766-f012:**
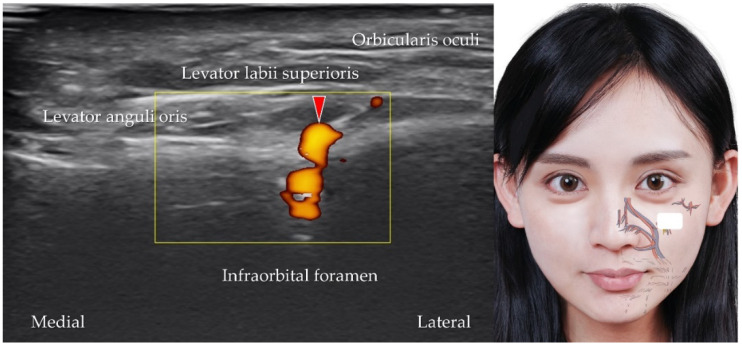
Power Doppler ultrasound imaging of the infraorbital artery (*red arrowhead*). *White square*, footprint of the ultrasound transducer.

**Figure 13 diagnostics-12-01766-f013:**
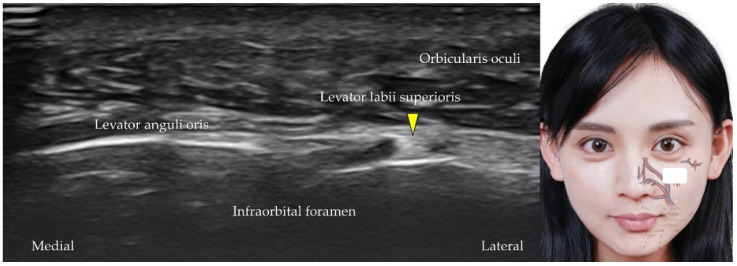
Ultrasound imaging of the infraorbital nerve (*yellow arrowhead*). *White square*, footprint of the ultrasound transducer.

**Figure 14 diagnostics-12-01766-f014:**
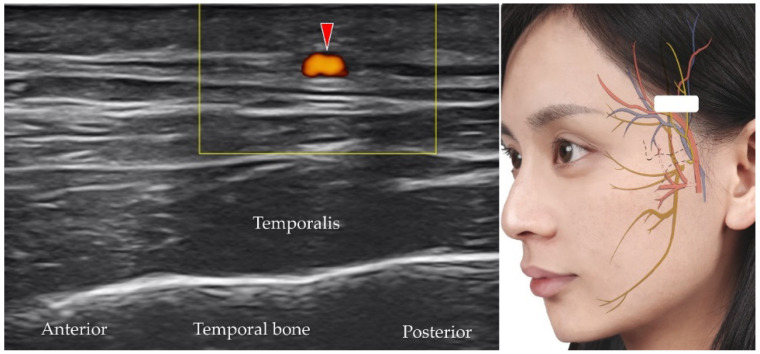
Power Doppler ultrasound imaging of the parietal branch of the superficial temporal artery (*red arrowhead*). *White square*, footprint of the ultrasound transducer.

**Figure 15 diagnostics-12-01766-f015:**
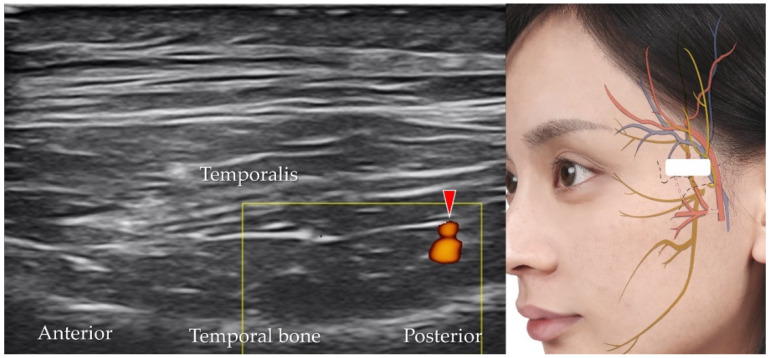
Power Doppler ultrasound imaging of the parietal branch of the deep temporal artery (*red arrowhead*). *White square*, footprint of the ultrasound transducer.

**Figure 16 diagnostics-12-01766-f016:**
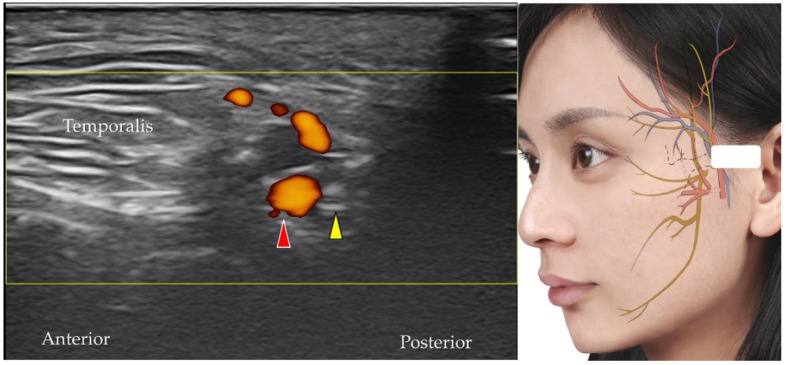
Power Doppler ultrasound imaging of the main trunk of the superficial temporal artery (*red arrowhead*) and the auriculotemporal nerve (*yellow arrowhead*). *White square*, footprint of the ultrasound transducer.

**Figure 17 diagnostics-12-01766-f017:**
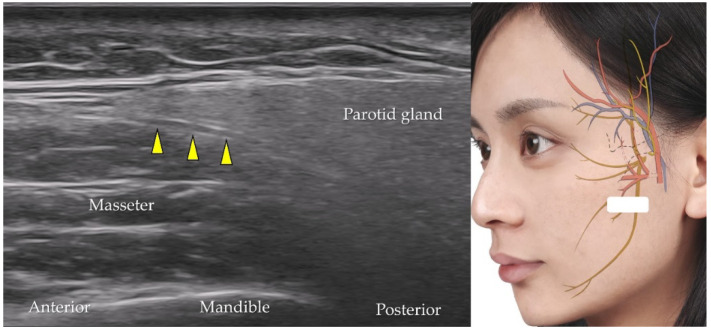
Ultrasound imaging of the facial nerve (*yellow arrowheads*). *White square*, the footprint of the ultrasound transducer.

**Figure 18 diagnostics-12-01766-f018:**
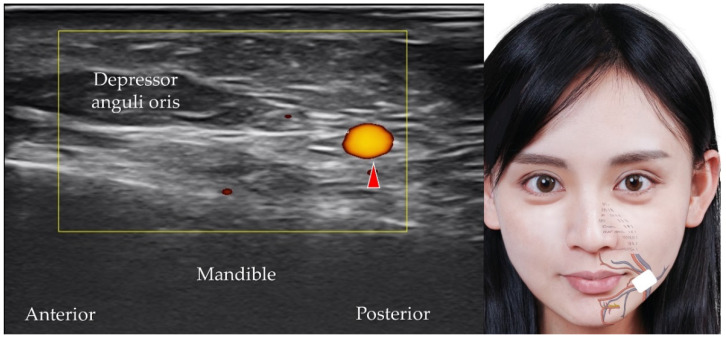
Power Doppler ultrasound imaging of the facial artery (*red arrowhead*). *White square*, footprint of the ultrasound transducer.

**Figure 19 diagnostics-12-01766-f019:**
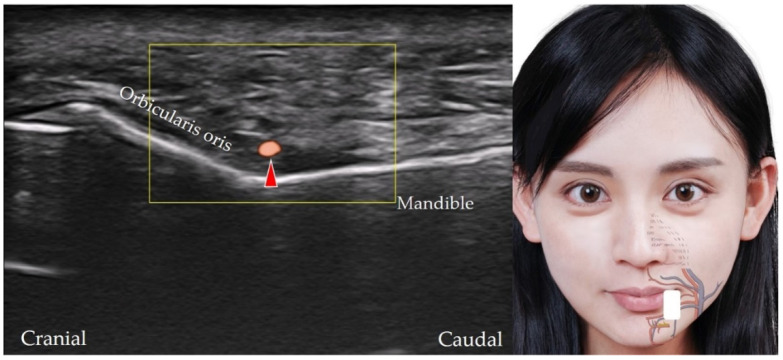
Power Doppler ultrasound imaging of the inferior labial artery (*red arrowhead*). *White square*, footprint of the ultrasound transducer.

**Figure 20 diagnostics-12-01766-f020:**
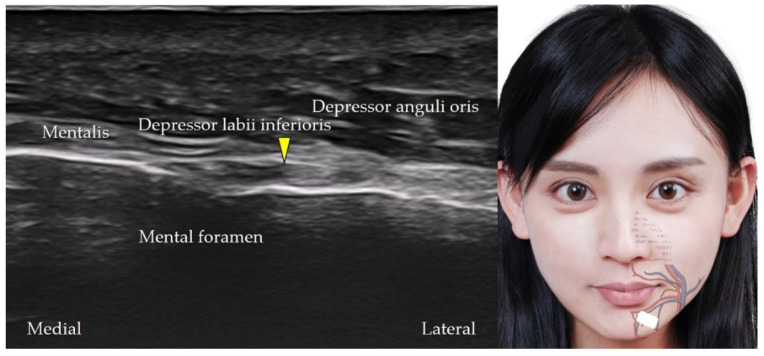
Ultrasound imaging of the mental nerve (*yellow arrowhead*). *White square*, footprint of the ultrasound transducer.

**Figure 21 diagnostics-12-01766-f021:**
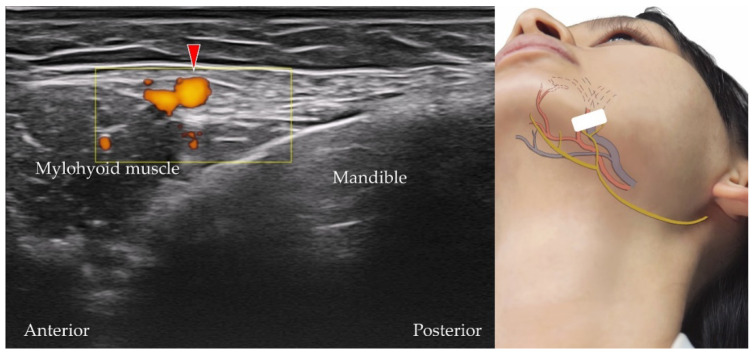
Power Doppler ultrasound imaging of the submental artery (*red arrowhead*). *White square*, footprint of the ultrasound transducer.

**Figure 22 diagnostics-12-01766-f022:**
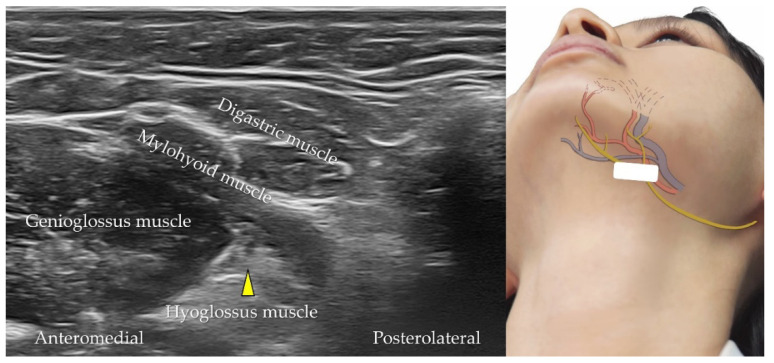
Ultrasound imaging of the hypoglossal nerve (*yellow arrowhead*). *White square*, footprint of the ultrasound transducer.

**Table 1 diagnostics-12-01766-t001:** Summary of the anatomy and scanning techniques for the neurovascular structures of the upper face.

Vessel/Nerve	Origin/Drain to	Supply	Transducer Position
Supratrochlear artery	Ophthalmic artery	Medial aspect of the upper eye lid forehead, glabella and frontalis muscle	The transducer is placed over the medial orbital rim in the horizontal plane
Supratrochlear vein	Supratrochlear foramen	Anterior forehead and scalp	The transducer is placed over the medial orbital rim in the horizontal plane
Supratrochlear nerve	Frontal nerve, branch of the ophthalmic nerve (V1)	Cutaneous innervation of the lateral lower forehead and upper eyelid	The transducer is placed over the medial orbital rim in the horizontal plane
Supraorbital artery	Ophthalmic artery	Upper eyelid and skin of the forehead and the scalp	The transducer is placed in proximity over the center of the superior orbital rim
Supraorbital vein	Angular vein	Forehead, eyebrow, and upper eyelid	The transducer is placed in proximity over the center of the superior orbital rim
Supraorbital nerve	Frontal nerve, branch of the ophthalmic nerve (V1)	Cutaneous sensation of the lateral forehead and upper eyelid	The transducer is placed in proximity over the center of the superior orbital rim
Frontal branch of the superficial temporal artery	Superficial temporal artery	Lateral aspect of the forehead with anastomosis of the supraorbital artery	The transducer is placed over the lateral part of the forehead slightly cranial to the eyebrow
Frontal branch of the superficial temporal vein	Superficial temporal vein	Lateral aspect of the forehead	The transducer is placed over the lateral part of the forehead slightly cranial to the eyebrow

**Table 2 diagnostics-12-01766-t002:** Summary of the anatomy and scanning techniques for the neurovascular structures over the middle face.

Vessel/Nerve	Origin/Drain to	Supply	Transducer Position
Dorsal nasal artery	Ophthalmic artery	Lacrimal sac and tip of the nose	The transducer is placed in the horizontal plane near the medial orbital rim
Dorsal nasal vein	Angular vein	Lateral side of the nose	The transducer is placed in the horizontal plane near the medial orbital rim
Angular artery	Facial artery	Lacrimal sac, nose, lower eyelid, and orbicularis oculi muscles	The transducer can be placed in the horizontal plane lateral to the ala of the nose
Angular vein	Facial vein	Medial canthus, and nose and upper lip	The transducer can be placed in the horizontal plane lateral to the ala of the nose lateral to the angular artery
Superior labial artery	Facial artery	Upper lip	The transducer is placed in the sagittal plane medial to the angle of the mouth
Superior labial vein	Facial vein	Upper lip	The transducer is placed in the sagittal plane medial to the angle of the mouth
Infraorbital artery	Maxillary artery	Lacrimal sac, upper incisor and canine teeth	The transducer is placed in the horizontal plane slightly distal to the inferior eyelid
Infraorbital vein	Pterygoid venous plexus	Lacrimal sac, upper incisor and canine teeth	The transducer is placed in the horizontal plane slightly distal to the inferior eyelid
Infraorbital nerve	Maxillary nerve (CN V2)	Lower eyelid, anterior cheek, and upper lip	The transducer is placed in the horizontal plane slightly distal to the inferior eyelid
Superficial temporal artery	External carotid artery	Parietal branch for the posterior temporal area and frontal branch for the lateral forehead	The transducer can be placed in the horizontal plane cranial to the zygomatic arch
Superficial temporal vein	Retromandibular vein	Posterior temporal area and lateral forehead	The transducer can be placed in the horizontal plane cranial to the zygomatic arch
Deep temporal artery	Maxillary artery	Temporalis muscle and temporal fossa	The transducer is placed in the horizontal plane cranial to the zygomatic arch to identify the vessels under the temporalis muscle
Deep temporal vein	Retromandibular vein	Temporalis muscle and temporal fossa	The transducer is placed in the horizontal plane cranial to the zygomatic arch to identify the vessels under the temporalis muscle
Auriculotemporal nerve	Mandibular nerve (CN V3)	Auricle, external acoustic meatus, and temporal region	The transducer is placed in the horizontal plane slightly anterior to the tragus
Facial nerve	Pons of the brain stem	Motor to facial expression muscles, the posterior belly of digastric, stylohyoid; sensory to taste (anterior two-thirds of tongue)	The transducer is placed in the horizontal plane at the cranial level of the mandibular ramus

**Table 3 diagnostics-12-01766-t003:** Summary of the anatomy and scanning techniques for the neurovascular structures over the lower face and submental region.

Vessel/Nerve	Origin/Drain to	Supply	Transducer Position
Lower face			
Facial artery	External carotid artery	Anterior upper neck and anterior mandible region	The transducer is placed in the horizontal plane at the midpoint of the mandibular body
Facial vein	Retromandibular or internal jugular vein	Anterior upper neck and anterior mandible region	The transducer is placed in the horizontal plane at the midpoint of the mandibular body
Inferior labial artery	Facial artery	Lower lip	The transducer is placed in the sagittal plane over the lower lip
Inferior labial Vein	Facial vein	Lower lip	The transducer is placed in the sagittal plane over the lower lip
Mental nerve	Mandibular nerve (CN V3)	Cutaneous sensation of the anterior chin and lower lip	The transducer is placed in the horizontal plane caudal to the lower lip and lateral to the mental protuberance
Submental region			
Submental artery	Facial artery	Submental area	The transducer is placed in the coronal plane over the submental region
Hypoglossal nerve	Brain stem	Tongue muscle	The transducer is positioned in the coronal plane over the submental area

## Data Availability

Data are contained within the main text of the manuscript.

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
