# Peer review of "Ultrasound Imaging of Facial Vascular Neural Structures and Relevance to Aesthetic Injections: A Pictorial Essay"

_diagnostics, 2022, doi:10.3390/diagnostics12071766_

Round 1

Reviewer 1 Report

1. What is the contribution of this paper. What is the impact of this review. What is the main point of this review manuscript. 

2. This review paper only provide few papers. The author should increase the number of cited papers.

3. It is very hard to find the connection between introduction, discussion, and conclusion of the papers. The introduction should be improved and associated with the discussion and the conclusion.

Author Response

Reviewer 1

  1. What is the contribution of this paper? What is the impact of this review? What is the main point of this review manuscript? 

Response: We appreciate the kind comment from the reviewer. We would like to apologize for selecting the incorrect article type when we submitted the manuscript. The manuscript should be categorized as a pictorial essay instead of a review, just like an article published last year (https://pubmed.ncbi.nlm.nih.gov/34679532/).

In order to resolve the reviewer’s concern, we have revised our title as “Ultrasound Imaging of Facial Vascular Neural Structures and Relevance to Aesthetic Injections: A Pictorial Essay“. The article type has been changed to “essay” upon resubmission. The main point of the essay has been described “In this pictorial essay, we demonstrated the sonoanatomy of the neurovascular structures in the facial and submental regions and its relevance to aesthetic injections, aiming to facilitate pre-procedure ultrasound scanning in a systematic manner”(line 49-52). The main contribution or impact has been stated as “We hope the accidental injury to the vessels and nerves on the face can be thus decreased during aesthetic injections” (line 52-53).    

  1. This review paper only provide few papers. The author should increase the number of cited papers.

Response: We appreciate the kind comments from the reviewer. The following articles have been cited in the revised manuscript for elaborating the sonoanatomy:

  1. Zhou, H.; Xue, Y.; Liu, P. Application of auriculotemporal nerve block and dextrose prolotherapy in exercise therapy of TMJ closed lock in adolescents and young adults. Head & Face Medicine 2021, 17, 11, doi:10.1186/s13005-021-00261-7.
  2. Cotofana, S.; Velthuis, P.J.; Alfertshofer, M.; Frank, K.; Bertucci, V.; Beleznay, K.; Swift, A.; Gavril, D.L.; Lachman, N.; Schelke, L. The Change of Plane of the Supratrochlear and Supraorbital Arteries in the Forehead-An Ultrasound-Based Investigation. Aesthet Surg J 2021, 41, Np1589-np1598, doi:10.1093/asj/sjaa421.
  3. Karahaliou, M.; Vaiopoulos, G.; Papaspyrou, S.; Kanakis, M.A.; Revenas, K.; Sfikakis, P.P. Colour duplex sonography of temporal arteries before decision for biopsy: a prospective study in 55 patients with suspected giant cell arteritis. Arthritis Res Ther 2006, 8, R116, doi:10.1186/ar2003.
  4. Tawfik, E.A.; Walker, F.O.; Cartwright, M.S. Neuromuscular ultrasound of cranial nerves. J Clin Neurol 2015, 11, 109-121, doi:10.3988/jcn.2015.11.2.109.
  5. Lee, K.L.; Lee, H.J.; Youn, K.H.; Kim, H.J. Positional relationship of superior and inferior labial artery by ultrasonography image analysis for safe lip augmentation procedures. Clin Anat 2020, 33, 158-164, doi:10.1002/ca.23379.
  6. Koziej, M.; Wnuk, J.; Polak, J.; Trybus, M.; Pękala, P.; Pękala, J.; Hołda, M.; Antoszewski, B.; Tomaszewski, K. The superficial temporal artery: A meta-analysis of its prevalence and morphology. Clin Anat 2020, 33, 1130-1137, doi:10.1002/ca.23550.
  7. Ren, H.; Shen, Y.; Luo, F. Treatment of Supraorbital Neuralgia Using Ultrasound-Guided Radiofrequency Thermocoagulation of the Supraorbital Nerve: A Retrospective Study. J Pain Res 2020, 13, 251-259, doi:10.2147/jpr.S228720.
  8. Alfertshofer, M.G.; Frank, K.; Moellhoff, N.; Helm, S.; Freytag, L.; Mercado-Perez, A.; Hargiss, J.B.; Dumbrava, M.; Green, J.B.; Cotofana, S. Ultrasound Anatomy of the Dorsal Nasal Artery as it Relates to Liquid Rhinoplasty Procedures. Facial Plast Surg Clin North Am 2022, 30, 135-141, doi:10.1016/j.fsc.2022.01.002.
  9. Michalek, P.; Donaldson, W.; McAleavey, F.; Johnston, P.; Kiska, R. Ultrasound imaging of the infraorbital foramen and simulation of the ultrasound-guided infraorbital nerve block using a skull model. Surg Radiol Anat 2013, 35, 319-322, doi:10.1007/s00276-012-1039-3.
  10. Yoshimatsu, H.; Harima, M.; Iida, T.; Narushima, M.; Karakawa, R.; Nakatsukasa, S.; Yamamoto, T.; Hayashi, A. Use of the Distal Facial Artery (Angular Artery) for Supermicrosurgical Midface Reconstruction. Plast Reconstr Surg Glob Open 2019, 7, e1978, doi:10.1097/gox.0000000000001978.

  1. It is very hard to find the connection between introduction, discussion, and conclusion of the papers. The introduction should be improved and associated with the discussion and the conclusion.

Response: We apologize again for not selecting the correct article category during submission. This draft should be treated as a pictorial essay, not a review. Therefore, there should be no session of discussion. In the introduction, we have highlighted that this is a pictorial essay, instead of a review. In this manner, we believe the introduction and conclusion will be more connected in the revised manuscript.

Reviewer 2 Report

This paper does not live up to the expectations provoked by the title. There is really not much information being given about nerves, not in the text, but also not in the figures. The paper does however give a nice and reasonably sound overview of ultrasound anatomy regarding the different arteries and veins. The information is not new, but given in a very readable manner. The title should be changed to relate more of the content being an overview of ultrasound and vessels rather than nerves.

I have the following remarks about the content of this paper: There is somewhat sloppy use of terms, e.g. the term injectate is used several times. This should better be named injectables. I have marked all these words yellow in the document. The structures indicated by arrows in certain figures are sometimes not correct or not formatted identically, e.g. ‘N.’  for nerve on Fig 1, whereas the others are depicted as ‘ nerve’. Wrong are terms in fig 2; fig 4 ‘ supratrochlear foramen is not clearly outlined. The term ‘ cosmetologists’  should be changed into ‘ injectors’.

The terms superior and inferior should be changed into cranial and caudal. Clearly wrong is Line 93-94 Page 4. Injection into a fascia should be avoided. Rephrasing is needed here.

The part about the angular vein (page 8 lines 187-192 + table 2) are incorrect. The angular artery and vein do not run together in that region. The vein is much more lateral than the artery.

Author Response

Reviewer 2

This paper does not live up to the expectations provoked by the title. There is really not much information being given about nerves, not in the text, but also not in the figures. The paper does however give a nice and reasonably sound overview of ultrasound anatomy regarding the different arteries and veins. The information is not new, but given in a very readable manner. The title should be changed to relate more of the content being an overview of ultrasound of vessels rather than nerves.

Response: We appreciate the kind comment from the reviewer. We agree with the reviewer that the vascular portion outweighs the nerve portion in the present manuscript. In compliance with the reviewer’s suggestion, we have changed our title as “Ultrasound Imaging of Facial Vascular Neural Structures and Relevance to Aesthetic Injections: A Pictorial Essay”. The term “vascular” is put ahead of “neural”, implying an overview of vessels more than nerves.

I have the following remarks about the content of this paper: There is somewhat sloppy use of terms, e.g. the term injecctate is used several times. This should better be named injectables. I have marked all these words yellow in the document. The structures indicated by arrows in certain figures are sometimes not correct or not formatted identically, e.g. ‘N.’  for nerve on Fig 1, whereas the others are depicted as ‘nerve’. Wrong are terms in fig 2; fig 4 ‘ supratrochlear foramen is not clearly outlined. The term ‘cosmetologists’ should be changed into ‘ injectors’.

Response: We appreciate the kind comments from the reviewer. In the system, we did not find the marked document. However, we did revise according to the reviewer’s suggestion. If the reviewer finds anything we do not revise, please do not hesitate to inform us.

First, the term “injecctables” has been revised as “injectables” (line 36 and 76) as suggested.

Second, in Figure 1, N. has been changed to “nerve”.

Third, in Figure 2, “deep temporal vessel” has been revised as “deep temporal artery”.

Fourth, the border of the supratrochlear foramen has been marked by the dashed in the revised manuscript.

Fifth, the term “cosmetologists” have been changed to “injectors”(line 309-310; line 317 and line 321).

The terms superior and inferior should be changed into cranial and caudal. Clearly wrong is Line 93-94 Page 4. Injection into a fascia should be avoided. Rephrasing is needed here.

Response:

We appreciate the kind comment from the reviewer. The sentence has been rephrased as “Subzygomatic depression can be corrected through injections into the superifical fat of the subcutaneous layer above the superficial musculoaponeurotic system distal to the zygomatic arch “ (line 94-96). The terms “superior” and “inferior” have all been changed to “carinal” and “caudal” if they do not belong to the anatomic structures such as “superior labial artery” or “inferior labial artery”.

The part about the angular vein (page 8 lines 187-192 + table 2) are incorrect. The angular artery and vein do not run together in that region. The vein is much more lateral than the artery.

Response: We appreciate the kind comments from the reviewer. We have revised the reciprocal positions between the angular artery and vein. In the revised manuscript, we have stated that the angular vein is more lateral than the artery as “The angular vein is seen more lateral than the angular artery“(line 193-194).

Reviewer 3 Report

In general I find the article interesting, just as a suggestion could include a comparison table of the devices used. 

Author Response

Review 3

In general, I find the article interesting, just as a suggestion could include a comparison table of the devices used. 

Response: We appreciate the kind comments from the reviewer. However, the only device used for drafting this pictorial essay is the ultrasound machine. The detail of the ultrasound machine is provided as “All sonographic images were obtained using a 10–25 MHz high-frequency linear transducer (X-Cube 90, Alpinion Medical Systems Co. Ltd., Anyang, Korea)”(line 56-58). Therefore, we are not able to provide a comparison table of the devices used. We appreciate the kind understanding of the reviewer.

Round 2

Reviewer 1 Report

The article had been revised properly. The authors showed a high effort to address the revised.